# Relationship Between Retinal Vascular Measurements and Anthropometric Indices in Patients Diagnosed with Persistent COVID-19

**DOI:** 10.3390/jcm14217857

**Published:** 2025-11-05

**Authors:** Rosario Alonso-Domínguez, Teresa Vicente-García, Silvia Arroyo-Romero, Nuria Suárez-Moreno, Alicia Navarro-Cáceres, Andrea Domínguez-Martín, Leticia Gómez-Sánchez, Cristina Lugones-Sánchez, Luis García-Ortiz, Alicia Ortega, Marta Gómez-Sánchez, Elena Navarro-Matias, Manuel A. Gómez-Marcos

**Affiliations:** 1Facultad de Enfermería y Fisioterapia, Universidad de Salamanca, Avenida Donantes de Sangre s/n, 37008 Salamanca, Spain; tvicente@usal.es (T.V.-G.); crislugsa@gmail.com (C.L.-S.); 2Primary Care Research Unit of Salamanca (APISAL), Institute of Biomedical Research of Salamanca (IBSAL), 37005 Salamanca, Spain; silvia_ar@usal.es (S.A.-R.); alicia.nav@usal.es (A.N.-C.); enavarro@saludcastillayleon.es (E.N.-M.);; 3Castilla and León Health Service-SACYL, Regional Health Management, 37005 Salamanca, Spain; 4Emergency Service, University Hospital of La Paz, P. of Castellana, 261, 28046 Madrid, Spain; 5Research Network on Chronicity, Primary Care and Health Promotion (RICAPPS), 37005 Salamanca, Spain; 6Department of Biomedical and Diagnostic Sciences, University of Salamanca, 37007 Salamanca, Spain; 7CIBER of Respiratory Diseases (CIBERES), Institute of Health Carlos III, 28029 Madrid, Spain; 8Group for Biomedical Research in Respiratory Infection & Sepsis (BioSepsis), Instituto de Investigación Biomédica de Salamanca (IBSAL), Gerencia Regional de Salud de Castilla y León, 37007 Salamanca, Spain; 9Home Hospitalization Service, Marques of Valdecilla University Hospital, s/n, 39008 Santander, Spain; 10Department of Medicine, University of Salamanca, 37008 Salamanca, Spain

**Keywords:** obesity, microvessels, retina, post-acute COVID-19 syndrome

## Abstract

**Introduction:** Persistent COVID-19 is associated with microvascular dysfunction, with retinal vessels as potential early biomarkers. Obesity, particularly visceral adiposity, contributes to this dysfunction; however, the body mass index (BMI) is limited in its ability to assess it. Therefore, more precise alternative anthropometric indices have been proposed, although their relationship with retinal vascular caliber in persistent COVID-19 has not been studied. **Objective:** To analyze the relationship between different anthropometric measurements and the caliber of retinal vessels in an adult population with persistent COVID-19. **Materials and Methods:** This was an observational, descriptive, and cross-sectional study and included individuals diagnosed with persistent COVID-19. Retinal images were obtained using a non-mydriatic retinograph. The anthropometric variables used included: waist and hip circumference, BMI, waist-to-height ratio (WHtR), Body Roundness Index (BRI), Abdominal Volume Index (AVI), and body composition parameters measured by bioelectrical impedance analysis. **Results:** The sample included 284 participants (mean age: 52.7 years; 31.8% men). Men exhibited greater general and abdominal adiposity. The AV Index was negatively associated with various anthropometric indicators (BMI, BRI, waist circumference, and AVI), while venular caliber showed positive associations with all these indices, except for BMI (*p* < 0.05 for all). No significant correlations were found between anthropometric values and arteriolar caliber. These associations persisted after adjusting for age, sex, and pharmacological treatment. **Conclusions:** Individuals with obesity are associated with alterations in retinal vessels in patients with persistent COVID-19, evidenced by an increase in venous caliber and a decrease in the AV Index. However, these findings should be interpreted cautiously.

## 1. Introduction

The COVID-19 pandemic has left a significant clinical aftermath in a considerable proportion of patients who, after acute infection, continue to experience symptoms for weeks or even months. This condition, known as long COVID or post-acute sequelae of SARS-CoV-2 infection, is characterized by multisystemic manifestations including fatigue, dyspnea, cognitive impairment, and cardiovascular and neurosensory disturbances [1]. In this context, the involvement of the vascular system, particularly at the microvascular level, has attracted increasing interest due to its potential role in the underlying pathophysiology of these prolonged symptoms [2].

The retinal microvascular circulation can be non-invasively accessed through imaging techniques [3] to provide a unique window into the status of systemic microcirculation [4]. Current evidence shows that alterations in the caliber of retinal arteries and veins are associated with hypertension, diabetes, dyslipidemia, cerebrovascular diseases, and vascular aging, among other conditions [5,6]. Consequently, assessing retinal vessel caliber may represent an early and sensitive biomarker of vascular damage in individuals with long COVID, as it is well-established that endothelial dysfunction affects the microcirculation much earlier than the macrocirculation in the course of vascular disease [7,8].

Among the various factors influencing vascular health, obesity holds a prominent place. The presence of obesity contributes to atherosclerosis by promoting fat accumulation in the walls of blood vessels, leading to intimal thickening, impaired blood flow, and vascular dysregulation, which may also impact retinal vascularization [9]. Traditionally, body mass index (BMI) has been the most widely used tool to estimate general obesity, and several studies have investigated the association between BMI and retinal vascular caliber [10,11]. However, the clinical utility of this index is limited as it does not differentiate between fat mass and lean mass, nor does it provide information about fat distribution [12]. This lack of precision has led to poorly defined categories such as individuals with “metabolically healthy obesity” or those with normal weight but high visceral fat, both presenting an increased risk of vascular disease not detected by BMI [13].

Given these limitations, several alternative anthropometric indices have been proposed to provide a more accurate assessment of cardiometabolic risk. Measures such as waist circumference, waist-to-hip ratio, waist-to-height ratio (WHtR), Body Roundness Index (BRI), and Abdominal Volume Index (AVI) have shown greater efficacy in identifying individuals with central obesity, systemic inflammation, and elevated cardiovascular risk, even when BMI is within the normal range [12,14]. These indices allow for a finer evaluation of visceral adipose tissue, which appears to play a key role in microvascular dysfunction [15].

Therefore, the aim of this study is to analyze the relationship between different anthropometric measures and retinal vessel caliber in an adult population with long COVID.

## 2. Materials and Methods

### 2.1. Study Design and Setting

This was an observational, descriptive, and cross-sectional study involving participants recruited as part of the BioICOPER study [16], which is registered on ClinicalTrials.gov (NCT05819840). The study was conducted at the Primary Care Research Unit of Salamanca (APISAL).

### 2.2. Study Population

Of the 798 individuals who initially attended Primary Care consultations and the Long COVID Unit, 305 met the World Health Organization (WHO) clinical definition of long COVID [1], agreed to participate, and fulfilled the inclusion criteria for the BioICOPER study. From these, 21 were excluded from the present analysis due to unassessable retinal images, resulting in 284 participants analyzed in this manuscript. Figure 1 displays the flow diagram of the participants included in this analysis.

These participants were consecutively recruited from Primary Care records of Salamanca and the Long COVID Unit of the Internal Medicine Department in the Salamanca Health Area. Exclusion criteria included: terminal illness, inability to attend the health center, a history of cardiovascular disease (ischemic heart disease or cerebrovascular disease), or an estimated glomerular filtration rate below 30%.

Participants did not take part in the study design. However, they were actively involved in the recruitment process, helping to disseminate the study’s objectives and inclusion criteria. Additionally, focus groups were conducted to explore their opinions and experiences regarding the impact of long COVID on vascular health.

At the end of the study, each participant received a detailed report summarizing all the tests performed.

### 2.3. Variables and Measurement Instruments

Data were collected by three trained healthcare professionals following a previously published standardized protocol [16]. Data quality was controlled by an external investigator.

#### 2.3.1. Sociodemographic Variables and Cardiovascular Risk Factors

Age, sex, marital status, educational level, and employment status information was collected. Participants were also asked about their history of hypertension, dyslipidemia, and diabetes, as well as the use of medications for these conditions.

Systolic blood pressure (SBP) and diastolic blood pressure (DBP) were measured according to the recommendations of the European Society of Hypertension [17]. Three measurements were taken on the dominant arm, with the patient seated and after five minutes of rest, using a validated sphygmomanometer (model M10-IT, omron healthcare, Barcelona, Spain).

#### 2.3.2. Lifestyle Variables

Physical activity was subjectively assessed using the Global Physical Activity Questionnaire (GPAQ) [18] developed by the World Health Organization. This instrument allowed the collection of information on physical activity levels and time spent in sedentary behaviors during the previous week. The amount of physical activity was expressed in METs/minute/week.

The participants of the study were classified as current smokers or non-smokers (if they have never smoked or have not smoked in the last year).

#### 2.3.3. Determination of Biomarkers of Endothelial Damage

Blood samples (6 mL) were collected in 3K-EDTA tubes at the Salamanca Primary Care Research Unit (APISAL), centrifuged (10 min, 2500 rpm), and plasma was stored at −20 °C until transfer to the BioSepsis Laboratory (University of Salamanca), where it was kept at −80 °C. Plasma concentrations of endothelial and inflammatory biomarkers (ICAM-1, VCAM-1, IL-6) were determined using the ELLA-SimplexPlex™ (Biotechne, Minneapolis, MN, USA) microfluidic immunoassay following the manufacturer’s instructions.

#### 2.3.4. Variables Related to Persistent COVID

From each participant’s medical history, the number and dates of COVID-19 infections, as well as symptoms related to long COVID were collected.

#### 2.3.5. Anthropometric Variables

Height was measured using a stadiometer (Seca 222, Birmingham, UK).

Weight was measured using the validated monitor InBody 230 (InBody Co., Ltd., Seoul, Republic of Korea) [19]. Measurements were taken in the morning with the participant barefoot, wearing light clothing, and having stood for approximately 5 min prior to the test, after at least 2 h of fasting and with an empty bladder.

Waist circumference was measured in millimeters using a flexible measuring tape. The measurement was taken by locating the upper edge of the iliac crests and wrapping the tape above this point without compressing the skin. According to the recommendations of the Spanish Society for the Study of Obesity (SEEDO) [20], abdominal obesity was defined as a waist circumference ≥ 102 cm in men and ≥88 cm in women.

Using these measurements, the following anthropometric indices were calculated:

BMI was calculated as weight divided by height in meters squared (kg/m^2^). Obesity was defined as BMI ≥ 30 kg/m^2^.

WHtR was calculated using the formula: WHtR = WC (cm)/height (cm).

BRI [21] was calculated using the formula: $BRI=364.2-365.5\sqrt{\left(1-{\left[(WC/2\pi )\right]}^{2}/{\left[(0.5\;height)\right]}^{2}\right)}$

AVI [22] was calculated using the formula: AVI = ((2 × (WC cm^2^) + 0.7 × (WC cm − height cm^2^))/1000.

Body composition was assessed via bioelectrical impedance analysis using the validated InBody 230 monitor [19], which estimates body fat mass (BFM), skeletal muscle mass (SMM), and fat-free mass (FFM). This device uses multiple frequencies and currents to perform a precise analysis of body composition without empirical estimation. For accurate measurements, the subject had to remain standing and clean the palms, thumbs, and soles of the feet with an InBody tissue before placing them properly on the monitor electrodes.

#### 2.3.6. Retinal Vessel Assessment

Retinal images were obtained using a non-mydriatic fundus camera (Topcon NW 200^®^, Topcon Europe BC, Capelle aan den IJssel, The Netherlands), capturing nasal and temporal views centered on the optic disc of both eyes. These images were analyzed using the AV Index Calculator software, developed by our research group (registration no. 00/2011/589) [23]. The software automatically detects the optic disc and traces two outer concentric circles to define two analysis zones: area A (0–0.5 cm in diameter) and area B (0.5–1 cm in diameter). It then identifies the edges of the blood vessels, distinguishes arteries from veins, and performs multiple measurements of the vessel diameters within area B.

Based on these measurements, the software estimates the mean caliber of arterioles and venules (in micrometers) and summarizes the results into three indicators: arterial thickness, venular thickness, and the arteriole-to-venule ratio (AV index). An AV index of 1.0 indicates that arteries and veins have on average, the same diameter, whereas values below 1 reflect narrower arteries.

To improve the reliability and efficiency of the analysis, only the main blood vessels in the superior and inferior temporal quadrants were considered, excluding the rest. Measurements were performed separately in each quadrant and then averaged to estimate the mean value for each eye [23]. Figure 2 illustrates the described process.

### 2.4. Statistical Analysis

The data were recorded using REDCap (Research Electronic Data Capture) [24].

Continuous variables are presented as means ± standard deviations, and categorical variables as frequencies and percentages. Mean comparisons based on sex were performed using Student’s *t*-test.

To analyze the association between retinal vessel caliber and the different anthropometric measurements, multiple regression analyses were conducted. The AV index and retinal vessel diameters were used as dependent variables; the anthropometric measurements as independent variables; and age (in years), sex (coded 0 = male, 1 = female), tobacco consumption (coded 1 = smoker, 0 = non-smoker), physical activity (in METs minute per week), and the use of antihypertensive, lipid-lowering, and glucose-lowering medications (coded 1 = consumer, 0 = non-consumer, for all) as covariates. All statistical analyses were performed using SPSS for Windows v28.0 (IBM Corp, Armonk, NY, USA).

Additionally, R software version 4.5.0 was used to analyze correlations between anthropometric parameters and retinal vessels. A heatmap was subsequently generated for graphical representation.

An alpha risk of 0.05 was set as the threshold for statistical significance in hypothesis testing.

### 2.5. Ethical Considerations

Following the recommendations of the Declaration of Helsinki [25], all participants were informed about the objectives of the study and signed an informed consent form prior to inclusion. In addition, the study was approved on 27 June 2022 by the Ethics Committee of Research with Medicines of the Health Area of Salamanca (CEIm Code: Ref. PI 2022 06 1048).

## 3. Results

### 3.1. Characteristics of the Study Population

The sample consisted of 284 participants, of whom 31.8% were male. The mean age was 52.71 ± 11.94 years. The time elapsed from the diagnosis of acute SARS-CoV-2 infection to inclusion in the study was 38.66 ± 9.58 months and the symptoms that were most frequently present were: fatigue 71.4%, weakness 67.4%, sleep disturbances 60.5% and dyspnea 58.2%. Table 1 shows the general characteristics of the patients stratified by sex.

Table 2 displays the description of the main study variables, including retinal blood vessel and anthropometric values. The mean AV index was 0.77 ± 0.11, with no significant differences between sexes. The mean arterial caliber was 106.74 µm in men and 105.60 µm in women, with no statistically significant difference. The mean venular caliber was 140.20 ± 15.26 µm (143.07 µm in men and 138.69 µm in women; *p* < 0.05).

Regarding anthropometric values, 32.5% of subjects were classified as individuals with obesity, with a higher prevalence among men (45.4%) than women (26.4%) (*p* < 0.001). Significant sex differences were also found in WHtR (60.57 vs. 55.44; *p* < 0.001) and BRI (5.67 vs. 4.60; *p* < 0.001). In terms of abdominal obesity, men presented a greater waist circumference (104.34 cm) compared to women (88.99 cm) (*p* < 0.001), as well as a higher AVI (22.12 in men vs. 16.48 in women; *p* < 0.001).

Significant differences were also observed in body composition measured by bioimpedance. The mean FFM was 46.86 kg and was higher in men (57.78 kg) than in women (41.75 kg) (*p* < 0.001); the same pattern was seen for SMM (32.27 kg in men vs. 22.52 kg in women; *p* < 0.001).

### 3.2. Association Between Anthropometric Values and Retinal Blood Vessels

Figure 3 illustrates the correlations between anthropometric and retinal blood vessel values. The AV index showed a negative correlation with BMI (−0.159), BRI (−0.146), WC (−0.174), WHtR (−0.145), AVI (−0.174), BFM (−0.143), FFM (−0.158), and SMM (−0.159) (*p* < 0.05 for all). However, the mean arterial caliber showed no significant correlation with any of the anthropometric indices. Conversely, venular caliber was positively correlated with BMI (0.157), BRI (0.152), WC (0.191), AVI (0.203), BFM (0.172), FFM (0.156), and SMM (0.155) (*p* < 0.05 for all values).

Multiple regression analysis adjusted for age, sex, tobacco consumption, physical activity and use of antihypertensive, lipid-lowering, and oral antidiabetic medications is shown in Table 3. The AV index was negatively associated with BMI (β = −0.003), BRI (β = −0.008), WC (β = −0.001), WHtR (β = −0.002), and AVI (β = −0.003), indicating that higher levels of adiposity were related to a lower (i.e., less favorable) AV Index. In contrast, venular caliber showed a positive association with BRI (β = 1.076), WC (β = 0.159), AVI (β = 0.462), and BFM (β = 0.195).

## 4. Discussion

This study analyzed the relationship between various anthropometric indices and retinal vessel caliber in a cohort of patients diagnosed with long COVID. Our results show that the AV index is negatively associated with several indicators of general and central adiposity, while venular caliber is positively associated with the same parameters. However, no significant associations were observed between anthropometric indices and retinal arterial caliber.

These findings are consistent with previous research that has identified a link between obesity and structural changes in the retinal microvasculature, particularly in venular caliber [11,26,27]. Excess adiposity, especially abdominal fat accumulation, has been associated with increased venular diameter, interpreted as a reflection of a chronic pro-inflammatory state [28]. Specifically, several inflammatory markers have been associated with venular dilation in the retina [29]. Our results expand this body of evidence by identifying such associations in a population with long COVID, a condition characterized by persistent inflammation and potential long-term endothelial dysfunction [30], which may partly explain the observed microvascular changes.

Furthermore, the association between obesity and increased venular caliber may also stem from the fact that individuals with obesity tend to have a higher total blood volume, which could lead to venous vessel dilation to accommodate the increased volume [31]. Finally, another mechanism that may explain this relationship is the elevated leptin concentrations found in individuals with obesity. Leptin is a hormone primarily secreted by adipose tissue and is capable of inducing both direct and indirect vasodilation through endothelial mechanisms, resulting in venular widening [32].

These findings may have clinical implications, as several studies suggest that retinal vascular diameter can improve through lifestyle changes, particularly those aimed at weight loss [33], as well as anti-inflammatory interventions [29]. This suggests a certain degree of plasticity in the human microvasculature when treatments are applied early in the course of disease, potentially mitigating the adverse effects associated with obesity on the systemic microvasculature.

On the other hand, the absence of a significant association between anthropometric indices and arterial caliber has been reported in other studies [34,35]. This reinforces the hypothesis that obesity more strongly affects the venous system of the retina than the arterial system, and that retinal arterial involvement is primarily mediated by arteriolar narrowing resulting from structural adaptations of the endothelium and vessel wall to chronic hypertension [36]. However, some studies have reported negative associations between obesity and arterial caliber [8,10].

Evidence linking obesity to the AV index remains limited. Nevertheless, a study conducted by Hanssen in children [37] aligns with our findings, as it showed an association between a reduced AV index and higher BMI, waist circumference, and other indicators of central adiposity in children with obesity. This observation may be related to the previously discussed venular dilation.

To date, there is no published scientific evidence evaluating retinal vessel thickness in individuals diagnosed simultaneously with long COVID and obesity. In this context, our study’s findings may be explained by an increase in retinal venous diameter, possibly related to endothelial dysfunction rather than a subclinical inflammatory component. In this regard, the values of the measured inflammatory markers (IL-6) were similar to those reported in a meta-analysis of health adults [38]. However, the values of one of the endothelial damage markers assessed, VCAM-1, were higher than those reported in healthy populations [39], supporting the hypothesis of an underlying endotheliopathy in individuals with long COVID.

In line with this interpretation, recent studies using advanced imaging techniques have documented persistent retinal microvascular abnormalities following SARS-CoV-2 infection. Several studies have demonstrated reduced capillary perfusion and a lower vessel density in the deep capillary plexus in convalescent and long COVID patients [40]. Moreover, impaired retinal oxygen metabolism a perfusion beyond the acute phase have been reported, suggesting sustained endothelial dysfunction and compromised tissue oxygenation [41]. Other cross-sectional studies have described microvascular loss and structural changes compatible with permanent capillary damage in severe or prolonged cases [42]. This findings reinforce the potential of retinal imaging as a non-invasive tool to detect microvascular and metabolic alterations after COVID-19 and justify the development of longitudinal studies to clarify whether the obesity-related associations observed in our work are specific to long COVID or reflect a broader microvascular vulnerability associated with excess adiposity.

Additionally, although the results are statistically significant, the relatively low R^2^ values observed in our models suggest that other unmeasured factors may contribute to the variability in retinal vessels caliber, indicating that obesity and long COVID are not the sole determinants of these microvascular changes.

Among the strengths of this study is the use of a broad set of anthropometric indices, including not only BMI but also more specific measures of central adiposity such as BRI and AVI. The analysis was also adjusted for key confounding variables (age, sex, and usual medication). Furthermore, a standardized technique was used to measure retinal vascular caliber, adding robustness to the results, and the study analyzed a previously unstudied population group.

Finally, some limitations must be acknowledged. The cross-sectional design precludes establishing causal relationships between anthropometric indices and the observed microvascular changes. Moreover, the absence of a control group without a history of COVID-19 infections prevents the determination of whether the observed associations are specific to long COVID or reflect broader relationships between obesity and microvascular health. Additionally, the study design does not allow for the formation of conclusions about the directionality of these associations, i.e., whether obesity contributes to microvascular alterations in the context of persistent COVID-19, whether long COVID exacerbates preexisting vascular changes, or whether both conditions share underlying mechanisms. Future research should consider longitudinal studies to evaluate the progression of microvascular alterations in relation to changes in body composition and inflammatory status in patients with long COVID. Moreover, retinal imaging could represent a valuable, non-invasive tool that provides a direct view of small-caliber vessels to monitor microvascular health in this population, particularly among obese individuals. This approach may help detect early vascular alterations and contribute to a better understanding of the systemic microcirculatory impact of long COVID.

## 5. Conclusions

In conclusion, within this cohort of patients with long COVID, indices of adiposity are associated with alterations in retinal vessels, particularly with increased venular caliber and a reduced AV index. However, these findings should be interpreted cautiously as it remains uncertain whether such associations are unique to individuals with long COVID or reflect more general effects of obesity on microvascular structure. Further longitudinal and comparative studies are warranted to clarify these relationships.

## Figures and Tables

**Figure 1 jcm-14-07857-f001:**
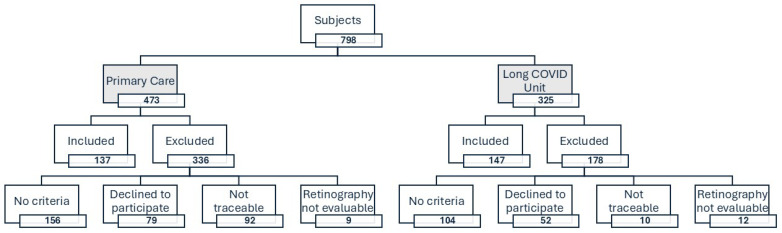
Flowchart of the study population.

**Figure 2 jcm-14-07857-f002:**
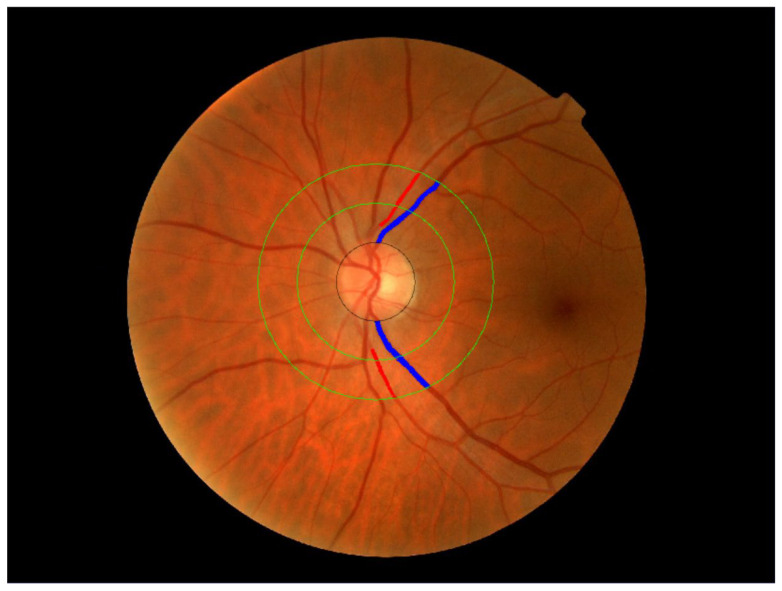
Method for measuring retinal vessel diameter.

**Figure 3 jcm-14-07857-f003:**
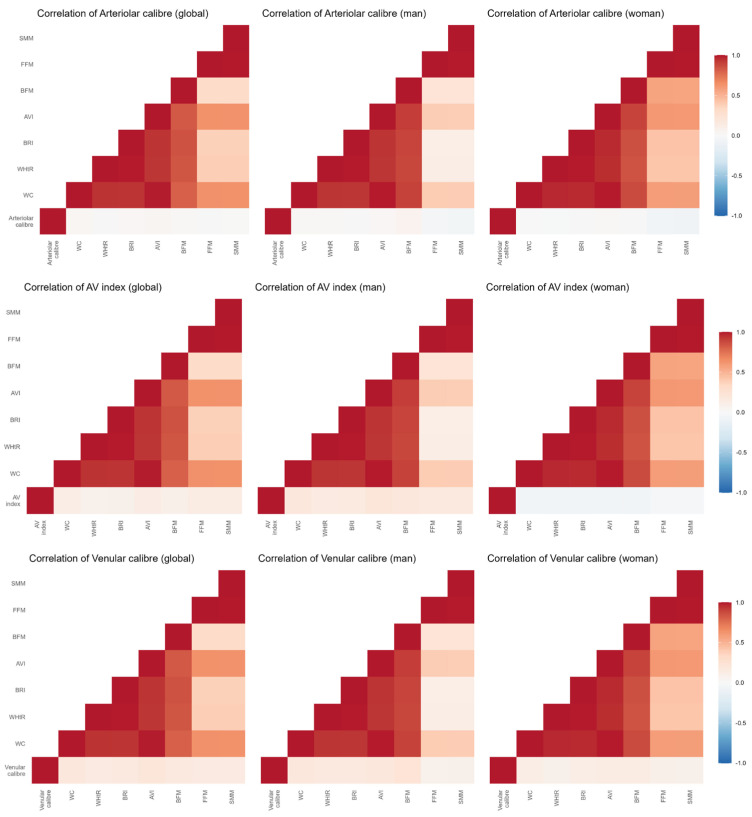
Correlation heatmap between anthropometric values and retinal blood vessels.

**Table 1 jcm-14-07857-t001:** General characteristics of the overall study sample and groups based on sex.

	Global (n = 284)		Men (n = 88)		Women (n = 196)		
	Mean or n	SD or (%)	Mean or n	SD or (%)	Mean or n	SD or (%)	*p*-Value
Conventional risk factors							
Age, (years)	52.71	11.94	55.70	12.28	51.32	11.54	0.004
SBP, (mmHg)	120.10	16.86	129.92	14.48	115.52	15.94	<0.001
DBP, (mmHg)	76.85	11.11	82.34	11.04	74.30	10.20	<0.001
Antihypertensive drugs, n (%)	79	(26.0)	34	(35.1)	45	(21.7)	0.014
Hypertension, n (%)	110	(36.2)	53	(54.6)	57	(27.5)	<0.001
FPG, (mg/dL)	87.88	17.67	94.37	19.77	84.84	15.74	<0.001
Hypoglycemic drugs, n (%)	32	(10.5)	18	(18.6)	14	(6.8)	<0.001
Diabetes Mellitus, n (%)	37	(12.2)	22	(22.7)	15	(7.3)	<0.001
Cholesterol total, (mg/dL)	187.49	34.50	182.11	32.94	190.01	35.00	0.063
LDL cholesterol, (mg/dL)	112.87	31.06	113.59	32.12	112.53	30.62	0.782
HDL cholesterol, (mg/dL)	56.98	13.66	48.78	10.86	60.82	13.15	<0.001
Triglycerides, (mg/dL)	102.12	50.46	117.47	54.39	94.92	46.94	<0.001
Lipid-lowering drugs, n (%)	75	(24.8)	40	(41.7)	35	(17.0)	<0.001
Dyslipidemia, n (%)	172	(57.0)	67	(69.1)	105	(51.2)	0.003
Lifestyle variables							
Smoker, n (%)	119	(41.9)	46	(52.3)	73	(37.1)	0.165
Physical activity (METs/min/week)	5113.18	5003.27	5394.02	5208.08	4982.21	4912.18	0.623
Biomarkers of endothelial damage							
IL-6 (pg/mL)	2.14	1.99	2.57	2.34	1.94	1.77	0.027
ICAM-1 (ng/mL)	259.19	77.43	265.31	90.21	256.33	70.74	0.253
VCAM-1 (ng/mL)	537.32	166.15	57.37	197.54	520.32	146.71	0.279

Values are means and standard deviations for continuous data, and number and proportions for categorical data. SBP: systolic blood pressure; DBP: diastolic blood pressure; FPG: fasting plasma glucose; LDL-C: low-density lipoprotein cholesterol; HDL-C: high-density lipoprotein cholesterol; IL-6: Interleukin-6; ICAM-1: Intercellular Adhesion Molecule-1; VCAM-1: Vascular Cell Adhesion Molecule-1. *p* value: differences between men and women.

**Table 2 jcm-14-07857-t002:** Descriptive retinal blood vessel and anthropometric values.

	Global (n = 284)		Men (n = 88)		Women (n = 196)		
	Mean or n	SD or (%)	Mean or n	SD or (%)	Mean or n	SD or (%)	*p*-Value
AV Index	0.77	0.11	0.76	0.08	0.78	0.11	0.099
Arteriolar calibre	105.94	11.79	106.74	12.52	105.60	11.47	0.473
Venular calibre	140.2	15.26	143.07	15.85	138.69	14.84	0.031
General obesity							
Obesity, n (%)	99	(32.5)	44	(45.4)	55	(26.4)	<0.001
BMI, (Kg/m^2^)	27.97	(5.55)	29.6	(4.64)	27.21	(5.78)	<0.001
WHtR	57.07	(8.94)	60.57	(7.56)	55.44	(9.10)	<0.001
BRI	4.94	(1.96)	5.67	(1.75)	4.60	(1.96)	<0.001
Abdominal obesity							
Abdominal obesity, n (%)	147	(48.2)	49	(50.5)	98	(47.1)	0.580
Waist circumference (cm)	93.87	(15.48)	104.34	(12.52)	88.99	(14.28)	<0.001
AVI	18.27	(5.84)	22.12	(5.27)	16.48	(5.20)	<0.001
Impedance measurement							
BFM	29.00	(11.18)	29.80	(10.60)	28.63	(11.44)	0.402
FFM	46.86	(10.02)	57.78	(8.36)	41.75	(5.75)	<0.001
SMM	25.63	(6.06)	32.27	(5.02)	22.52	(3.44)	<0.001

Values are means and standard deviations for continuous data, and number and proportions for categorical data. BMI. Body mass index; WHtR: waist height ratio; BRI: Body roundness index; AVI: Abdominal volume index; BFM: body fat mass; FFM: fat-free mass; SMM: skeletal muscle mass. *p* value: differences between men and women.

**Table 3 jcm-14-07857-t003:** Multiple regression adjusted for age, sex, drugs, physical activity and tobacco consumption.

	β	IC (95%)	*p*-Value	R^2^
AV Index				
BMI	−0.003	−0.005–0.000	0.030	0.047
BRI	−0.008	−0.015–−0.001	0.029	0.048
WC	−0.001	−0.002–0.000	0.011	0.054
WHtR	−0.002	−0.003–0.000	0.032	0.047
AVI	−0.003	−0.006–0.001	0.012	0.053
BFM	−0.038	−0.179–0.103	0.595	0.021
FFM	−0.126	−0.350–0.098	0.270	0.024
SMM	−0.231	−0.607–0.145	0.227	0.025
Arteriolar calibre (µm)				
BMI	−0.154	−0.429–0.121	0.271	0.024
BRI	−0.190	−1.009–0.628	0.647	0.020
WC	−0.029	−0.141–0.082	0.603	0.020
WHtR	−0.056	−0.235–0.123	0.541	0.013
AVI	−0.054	−0.349–0.241	0.719	0.020
BFM	−0.038	−0.179–0.103	0.595	0.021
FFM	−0.126	−0.350–0.098	0.270	0.024
SMM	−0.231	−0.607–0.145	0.227	0.025
Venular calibre (µm)				
BMI	0.317	−0.031–0.665	0.074	0.052
BRI	1.076	0.043–2.109	0.041	0.056
WC	0.159	0.019–0.299	0.027	0.058
WHtR	0.212	−0.014–0.439	0.066	0.053
AVI	0.462	0.091–0.833	0.015	0.062
BFM	0.195	0.018–0.371	0.031	0.056
FFM	0.180	−0.103–0.463	0.211	0.045
SMM	0.302	−0.174–0.777	0.212	0.045

Multiple regression analysis using AV index, arteriolar caliber and venular caliber as dependent variables; BMI, BRI, WC, WHtR, AVI, BFM, FFM and SMM, as independent variables; and age, sex, tobacco consumption, physical activity and antihypertensive, antiglycemic and antilipidemic drugs as adjustment variables. BMI: Body mass index; WHtR: waist height ratio; BRI: Body roundness index; AVI: Abdominal volume index; BFM: body fat mass; FFM: fat-free mass; SMM: skeletal muscle mass.

## Data Availability

The data supporting the findings of this study are available on ZENODO under the DOI. 10.5281/zenodo.14282873.

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
