# Peer review of "Relationship Between Retinal Vascular Measurements and Anthropometric Indices in Patients Diagnosed with Persistent COVID-19"

_jcm, 2025, doi:10.3390/jcm14217857_

Round 1
Reviewer 1 Report
Comments and Suggestions for Authors
An interesting study i thank the authors for allowing me to review this paper i have couple of small suggestions:
In the methods section, elaborate on the consideration of additional confounders, such as smoking status or physical activity levels, which may influence vascular outcomes, or provide justification for their omission. The reason behind it is due to the fact that Table 3 results have low R^2, while remaining statistically significant, but the practical clinical relevance may be limited. It is recommended to acknowledge in the discussion that the low R-squared values indicate potential influence from unmeasured confounders.
Enhance the discussion by incorporating recent studies on retinal changes in long COVID, such as those demonstrating reduced capillary perfusion in recovered patients and impaired retinal oxygen metabolism and perfusion. This would strengthen the manuscript's relevance and contextualization.
Consider elaborating on clinical implications, such as the potential utility of retinal imaging as a non-invasive tool for monitoring microvascular health in obese patients with long COVID.
Inconsistent usage of "WHtC" (standardize to "WHtR" if intended)
Author Response
Reviewer #1:
- An interesting study i thank the authors for allowing me to review this paper i have couple of small suggestions:
Authors' Answer
We sincerely thank the reviewer for their careful reading of our manuscript and for their thoughtful comments and suggestions, which have helped us to improve the quality and clarity of the paper. We truly appreciate the positive evaluation and constructive feedback.
- In the methods section, elaborate on the consideration of additional confounders, such as smoking status or physical activity levels, which may influence vascular outcomes, or provide justification for their omission. The reason behind it is due to the fact that Table 3 results have low R^2, while remaining statistically significant, but the practical clinical relevance may be limited. It is recommended to acknowledge in the discussion that the low R-squared values indicate potential influence from unmeasured confounders.
Authors' Answer
We thank the reviewer for this observation. Following their recommendations, we have included tobacco use and physical activity (expressed in METs/min/week) as potential confounding factors in the multiple regression analysis. In the current version, the inclusion of these variables as covariates in regression models is detailed, and the corresponding results in Table 3 have been updated. Additionally, in the Discussion, it is noted that, although the results are statistically significant, the low R2 values may reflect the influence of other unmeasured factors contributing to the observed variability.
We have made changes to the methodology, results, and discussion sections of the manuscript and in Tables 1 and 3, as shown below:
Lines 126-133:
2.3.2. Lifestyle variables
Physical activity was subjectively assessed using the Global Physical Activity Questionnaire (GPAQ) [18], developed by the World Health Organization. This instrument allowed the collection of information on physical activity levels and time spent in sedentary behaviors during the previous week. The amount of physical activity was expressed in METs/minute/week.
The participants of the study will be classified as current smokers or non-smokers (if they have never smoked or have not smoked in the last year).
Lines 196-201: The AV index and retinal vessel diameters were used as dependent variables, the anthropometric measurements as independent variables, and age (in years), sex (coded 0= male, 1=female), tobacco consumption (coded 1=smoker, 0= non-smoker), physical activity (in METs minute per week), and the use of antihypertensive, lipid-lowering, and glucose-lowering medications (coded 1=consumer, 0= non-consumer, for all), as covariates.
Lines 262-268: Multiple regression analysis adjusted for age, sex, tobacco consumption, physical activity and use of antihypertensive, lipid-lowering, and oral antidiabetic medications is shown in Table 3. The AV index was negatively associated with BMI (β = –0.003), BRI (β = –0.008), WC (β = –0.001), WHtR (β = –0.002), and AVI (β = –0.003), indicating that higher levels of adiposity were related to lower (i.e., less favorable) AV Index. In contrast, venular caliber showed a positive association with BRI (β = 1.076), WC (β = 0.159), AVI (β = 0.462), and BFM (β = 0.195).
Lines 342-345: Additionally, although the results are statistically significant, the relatively low R² values observed in our models suggest that other unmeasured factors may contribute to the variability in retinal vessels caliber, indicating that obesity and long COVID are not the sole determinants of these microvascular changes.
- Enhance the discussion by incorporating recent studies on retinal changes in long COVID, such as those demonstrating reduced capillary perfusion in recovered patients and impaired retinal oxygen metabolism and perfusion. This would strengthen the manuscript's relevance and contextualization.
Authors' Answer
We have expanded the Discussion to include recent, relevant studies reporting retinal microvascular alterations after SARS-CoV-2 infection.
Lines 329-341: In line with this interpretation, recent studies using advanced imaging techniques have documented persistent retinal microvascular abnormalities following SARS-CoV-2 infection. Several studies have demonstrated reduced capillary perfusion a lower vessel density in the deep capillary plexus in convalescent and long COVID patients [40]. Moreover, impaired retinal oxygen metabolism a perfusion beyond the acute phase have been reported, suggesting sustained endothelial dysfunction and compromised tissue oxygenation [41]. Other cross-sectional studies have described microvascular loss and structural changes compatible with permanent capillary damage in severe or prolonged cases [42]. This findings reinforce the potential of retinal imaging as a non invasive-tool to detect microvascular and metabolic alterations after COVID-19 and justify the development of longitudinal studies to clarify whether the obesity-related associations observed in our work are specific to long COVID o reflect a broader microvascular vulnerability associated with excess adiposity.
- Consider elaborating on clinical implications, such as the potential utility of retinal imaging as a non-invasive tool for monitoring microvascular health in obese patients with long COVID.
Authors' Answer
Following their recommendation, we have expanded the final paragraph of the Discussion section to address the potential clinical implications of our findings.
Lines 363-367: Moreover, retinal imaging could represent a valuable, non-invasive tool that provides a direct view of small-caliber vessels, for monitoring microvascular health in this population, particularly among obese individuals. This approach may help detect early vascular alterations and contribute to a better understanding of the systemic microcirculatory impact of long COVID.
- Inconsistent usage of "WHtC" (standardize to "WHtR" if intended)
Authors' Answer
We thank the reviewer for noticing this inconsistency. We have carefully revised the manuscript and standardized the terminology, replacing all instances of “WHtC” with “WHtR” as intended.

Reviewer 2 Report
Comments and Suggestions for Authors
Dear editors and authors, I would like to thank you for the opportunity to review the manuscript. This manuscript presents a cross-sectional study that examined the relationship between various anthropometric measures and retinal vessel calibrations in a cohort of 284 patients with persistent COVID-19 (long-term COVID). The main conclusion is that the indicators of total and central obesity are negatively related to the arteriole-venule ratio index (AV) and positively related to the caliber of veins, but do not show a significant relationship with the caliber of arterioles. Much attention is paid to the long-term cohort of COVID patients. The article examines an insufficiently studied aspect of the syndrome - the state of microvessels — in a specific clinical context. At the same time, in their study, the authors go beyond the simplified body mass index (BMI) to include a reliable set of alternative indicators (WHtR, BRI, AVI) and body composition data obtained on the basis of bioimpedance analysis. This gives a more detailed understanding of obesity and its distribution, body components. The multiple regression models are adjusted accordingly to take into account key factors such as age, gender, and appropriate pharmacotherapy, which increases the reliability of the associations obtained. Сomments. The most significant limitation is the absence of a control group in which there were no cases of COVID-19 infection in the anamnesis or a long period of COVID. Thus, it is impossible to determine whether the observed relationships are unique to the long-term COVID population or whether they are increasing. The name and conclusions imply a specific link to persistent COVID-19, but the design of the study cannot confirm this. The authors should explicitly acknowledge this limitation during the discussion and moderate their conclusions accordingly. They should point out that although these associations exist within the long-term COVID cohort, it remains unknown whether they differ from the comparable control population. A phrase like "... in patients with persistent COVID-19" should be framed as a description of the test sample, and not as a statement about a unique pathophysiology. The cross-sectional nature precludes any conclusions about causality or orientation. It is impossible to determine whether obesity causes microvascular changes with prolonged COVID, whether prolonged COVID exacerbates the microvascular effects of pre-existing obesity, or whether both of these factors are influenced by a third, immeasurable factor. The study also lacks detailed information about the long-term COVID phenotype of the participants. Key information is missing, such as the time elapsed since acute infection, the severity of the initial infection, and a specific set of persistent symptoms (e.g., fatigue, shortness of breath, cognitive impairment). These factors can be critical hindrances or effect modifiers. During the discussion, it is reasonably hypothesized that chronic inflammation is a potential mechanism linking obesity and venous dilation in the long-term course of COVID. However, the study did not include any direct indicators of systemic inflammation (e.g. CRP, IL-6) or endothelial dysfunction (e.g. biomarkers such as VCAM-1, ICAM-1). The phrase "The AV index was negatively associated with various anthropometric indicators (BMI, BRI, waist circumference, and AVI)" is a bit confusing on first reading, as one would expect a positive relationship. Clarifying that a higher obesity index is associated with a lower (i.e., worse) AV index would provide more clarity. The flowchart (Figure 1) shows the initial set of 798 people, but the text states: "There are 305 people in total... they were consistently recruited." The relationship between these two numbers should be briefly explained in the title or text. Results, Table 1/2: The "p-vale" column header in Tables 1 and 2 contains a typo and should be corrected to "p-value." I recommend Major Revision. Before the manuscript is considered for publication, it is necessary to resolve the issues raised above, in particular, to soften the conclusions and discuss the limitations more thoroughly.
Author Response
Reviewer #2:
- Dear editors and authors, I would like to thank you for the opportunity to review the manuscript. This manuscript presents a cross-sectional study that examined the relationship between various anthropometric measures and retinal vessel calibrations in a cohort of 284 patients with persistent COVID-19 (long-term COVID). The main conclusion is that the indicators of total and central obesity are negatively related to the arteriole-venule ratio index (AV) and positively related to the caliber of veins, but do not show a significant relationship with the caliber of arterioles. Much attention is paid to the long-term cohort of COVID patients. The article examines an insufficiently studied aspect of the syndrome - the state of microvessels — in a specific clinical context. At the same time, in their study, the authors go beyond the simplified body mass index (BMI) to include a reliable set of alternative indicators (WHtR, BRI, AVI) and body composition data obtained on the basis of bioimpedance analysis. This gives a more detailed understanding of obesity and its distribution, body components. The multiple regression models are adjusted accordingly to take into account key factors such as age, gender, and appropriate pharmacotherapy, which increases the reliability of the associations obtained.
Authors' Answer
We sincerely thank the reviewer for their careful reading of our manuscript and for their detailed and constructive comments. We greatly appreciate the positive evaluation of our study design, the inclusion of multiple anthropometric and body composition measures, and the methodological rigor applied in the regression analyses. The reviewer’s insights are very valuable and encouraging for the improvement of our work.
- С The most significant limitation is the absence of a control group in which there were no cases of COVID-19 infection in the anamnesis or a long period of COVID. Thus, it is impossible to determine whether the observed relationships are unique to the long-term COVID population or whether they are increasing. The name and conclusions imply a specific link to persistent COVID-19, but the design of the study cannot confirm this. The authors should explicitly acknowledge this limitation during the discussion and moderate their conclusions accordingly. They should point out that although these associations exist within the long-term COVID cohort, it remains unknown whether they differ from the comparable control population. A phrase like "... in patients with persistent COVID-19" should be framed as a description of the test sample, and not as a statement about a unique pathophysiology. The cross-sectional nature precludes any conclusions about causality or orientation. It is impossible to determine whether obesity causes microvascular changes with prolonged COVID, whether prolonged COVID exacerbates the microvascular effects of pre-existing obesity, or whether both of these factors are influenced by a third, immeasurable factor.
Authors' Answer
We thank the reviewer for these insightful comments and fully agree with the need to clarify the interpretation of our findings. Following their suggestions, we have revised both the Limitations and Conclusions sections to explicitly acknowledge the absence of a control group and to moderate our statements regarding the specificity of the observed associations to long COVID.
Lines 352-360: Finally, some limitations must be acknowledged. The cross-sectional design precludes establishing causal relationships between anthropometric indices and the observed microvascular changes. Moreover, the absence of a control group without a history of COVID-19 infections prevents determining whether the observed association are specific to long COVID or reflect broader relationships between obesity and microvascular health. Additionally, the study design does not allow conclusion about the directionality of these associations, whether obesity contributes to microvascular alterations in the context of persistent COVID-19, whether long COVID exacerbates preexisting vascular changes, or whether both conditions share underlying mechanisms.
Lines 369-374: In conclusion, within this cohort of patients with long COVID, indices of adiposity are associated with alterations in retinal vessels, particularly with increased venular caliber and a reduced AV index. However, these findings should be interpreted cautiously, as it remains uncertain whether such associations are unique to individuals with long COVID or reflect more general effects of obesity on microvascular structure. Further longitudinal and comparative studies are warranted to clarify these relationships.
- The study also lacks detailed information about the long-term COVID phenotype of the participants. Key information is missing, such as the time elapsed since acute infection, the severity of the initial infection, and a specific set of persistent symptoms (e.g., fatigue, shortness of breath, cognitive impairment). These factors can be critical hindrances or effect modifiers.
Authors' Answer
We thank the reviewer for this comment. We have now included information on the time elapsed from COVID-19 diagnosis to study inclusion, as well as detailed description of the most frequent symptoms.
Lines 141-143:
2.3.4. Variables related to persistent COVID.
From each participant’s medical history, information was collected on the number and dates of COVID-19 infections, as well as symptoms related to long COVID.
.
Lines 217-220: The time elapsed from the diagnosis of acute SARS-CoV2 infection to inclusion in the study was 38.66 ± 9.58 months and the symptoms that were most frequently present were: fatigue 71.4%, weakness 67.4%, sleep disturbances 60.5% and dyspnea 58.2%.
- During the discussion, it is reasonably hypothesized that chronic inflammation is a potential mechanism linking obesity and venous dilation in the long-term course of COVID. However, the study did not include any direct indicators of systemic inflammation (e.g. CRP, IL-6) or endothelial dysfunction (e.g. biomarkers such as VCAM-1, ICAM-1).
Authors' Answer
We appreciate the reviewer’s observation. We have now included the corresponding values for IL-6, VCAM-1, and ICAM-1 in both the Methods section and Table 1. Additionally, we have revised the relevant section of the Discussion.
Lines 134-140:
2.3.3. Determination of biomarkers of endothelial damage
Blood samples (6 mL) were collected in 3K-EDTA tubes at the Salamanca Primary Care Research Unit (APISAL), centrifuged (10 min, 2,500 rpm), and plasma was stored at - 20 °C until transfer to the BioSepsis Laboratory (University of Salamanca), where it was kept at - 80 °C. Plasma concentrations of endothelial and inflammatory biomarkers (ICAM-1, VCAM-1, IL-6) were determined using the ELLA-SimplexPlex™ (Biotechne) microfluidic immunoassay, following the manufacturer’s instructions.
Lines 323-328: In this regard, the values of the measured inflammatory markers (IL-6) were similar to those reported in a meta-analysis of health adults [38]. However, the values of one of the endothelial damage markers assessed, VCAM-1, were higher than those reported in healthy populations [39], supporting the hypothesis of an underlying endotheliopathy in individuals with long COVID.
- The phrase "The AV index was negatively associated with various anthropometric indicators (BMI, BRI, waist circumference, and AVI)" is a bit confusing on first reading, as one would expect a positive relationship. Clarifying that a higher obesity index is associated with a lower (i.e., worse) AV index would provide more clarity.
Authors' Answer
Following the reviewer's recommendations we have included clarification.
Lines 264-266: The AV index was negatively associated with BMI (β = –0.003), BRI (β = –0.008), WC (β = –0.001), WHtR (β = –0.002), and AVI (β = –0.003), indicating that higher levels of adiposity were related to lower (i.e., less favorable) AV Index.
- The flowchart (Figure 1) shows the initial set of 798 people, but the text states: "There are 305 people in total... they were consistently recruited." The relationship between these two numbers should be briefly explained in the title or text.
Authors' Answer
Following the author’s recommendation, we have expanded the information in the methodology section as follows:
Lines 95-99: Of the 798 individuals who initially attended Primary Care consultations and the Long COVID Unit, 305 met the World Health Organization (WHO) clinical definition of long COVID [1], accepted to participate, and fulfilled the inclusion criteria for the BioICOPER study. From these, 21 were excluded from the present analysis due to unassessable retinal images, resulting in 284 participants analyzed in this manuscript.
- Results, Table 1/2: The "p-vale" column header in Tables 1 and 2 contains a typo and should be corrected to "p-value."
Authors' Answer
We thank the reviewer for noticing this typographical error. The column headers in Tables 1 and 2 have been corrected to “p-value”.
- I recommend Major Revision. Before the manuscript is considered for publication, it is necessary to resolve the issues raised above, in particular, to soften the conclusions and discuss the limitations more thoroughly.
Authors' Answer
We sincerely thank the reviewer for their careful evaluation and constructive suggestions. We have carefully addressed all the points raised, particularly by softening the conclusions and expanding the discussion of the study’s limitations as recommended. We believe these revisions have improved the clarity and balance of the manuscript.

Round 2
Reviewer 2 Report
Comments and Suggestions for Authors
Dear editors and authors of the manuscript, the revised manuscript demonstrates significant improvements in terms of clarity, scientific rigor, and contextualization of the conclusions it contains. The introduction now explains the limitations of body mass index more convincingly and provides a clear rationale for using alternative indicators such as the body roundness index and the abdominal volume index to better assess visceral obesity and its relationship to microvascular health. The discussion was reinforced by the clear connection of the results obtained with the known pathophysiology of COVID over a long period of time, in particular with persistent endothelial dysfunction and inflammation. This makes the study not just an observation, but an investigation into the potential mechanism underlying the symptoms of the syndrome. The results are presented in the form of special tables (Tables 1, 2) and a correlation heat map (Fig. 3), which provides a clear and multifaceted presentation of the data. The inclusion of both unadjusted correlations and adjusted regression coefficients increases the depth of analysis. A significant improvement is the inclusion of recent studies using advanced retinal imaging in COVID patients over a long period of time. The manuscript now presents the results of studies on reducing capillary perfusion, vascular density, and impaired oxygen metabolism, which places her own results in a broader, rapidly developing field and enhances the importance of retinal imaging. The Limitations section is reliable and critically meaningful. It correctly points to the inability of end-to-end analysis to establish a cause-and-effect relationship and the critical absence of a non-COVID-related control group. Such honesty allows readers to adequately evaluate the results obtained and indicates directions for future research. The conclusions are rather cautious, stating that the findings "should be interpreted with caution," and acknowledging the uncertainty about whether these associations are unique to COVID in the long term or a common consequence of obesity. These improvements transform the manuscript from a simple report of associations into a more impactful and scientifically robust contribution that effectively sets the stage for future research.